# Does Dietary Sodium Alginate with Low Molecular Weight Affect Growth, Antioxidant System, and Haemolymph Parameters and Alleviate Cadmium Stress in Whiteleg Shrimp (*Litopenaeus vannamei*)?

**DOI:** 10.3390/ani13111805

**Published:** 2023-05-30

**Authors:** Dara Bagheri, Rohullah Moradi, Mahyar Zare, Ebrahim Sotoudeh, Seyed Hossein Hoseinifar, Amin Oujifard, Noah Esmaeili

**Affiliations:** 1Faculty of Nano and Bio Science and Technology, Department of Fisheries, Persian Gulf University, Bushehr 75169, Iran; 2Faculty of Fisheries and Protection of Waters, South Bohemian Research Center of Aquaculture and Biodiversity of Hydrocenoses, Institute of Aquaculture and Protection of Waters, University of South Bohemia in České Budějovice, 370 05 České Budějovice, Czech Republic; 3Faculty of Fisheries and Environmental Sciences, Department of Fisheries, Gorgan University of Agricultural Sciences and Natural Resources, Gorgan 4913815739, Iran; 4Institute for Marine and Antarctic Studies, University of Tasmania, Hobart, TAS 7005, Australia; noah.esmaeili@utas.edu.au

**Keywords:** antioxidant system, serological enzyme, prebiotics, cadmium

## Abstract

**Simple Summary:**

This study investigated the effect of low molecular weight sodium alginate (LMWSA) on the growth and health of witheleg shrimp (*Litopenaeus vannamei*). Further, we tested whether LMWSA can alleviate the negative impacts of cadmium. This study showed that this additive could improve the feed conversion ratio (FCR) and antioxidant parameters. While cadmium suppressed the antioxidant system parameters in the control group, these parameters were not decreased in those fed dietary LMWSA.

**Abstract:**

Decreasing low molecular weight can improve the digestibility and availability of ingredients such as sodium alginate. This study aimed to test the four dosages of low molecular weight sodium alginate (LMWSA) (0%: Control, 0.05%: 0.5 LMWSA, 0.10%: 1.0 LMWSA, and 0.2%: 2.0 LMWSA) in whiteleg shrimp (*Litopenaeus vannamei*) (3.88 ± 0.25 g) for eight weeks. After finishing the trial, shrimp were exposed to cadmium (1 mg/L) for 48 h. While feed conversion ratio (FCR) improved in shrimp fed dietary 2.0 LMWSA (*p* < 0.05), there was no significant difference in growth among treatments. The results showed a linear relation between LMWSA level and FCR, and glutathione S-transferase (GST) before; and malondialdehyde (MDA), glutathione (GSH), GST, and alanine transaminase (ALT) after cadmium stress (*p* < 0.05). The GST, MDA, ALT, and aspartate transaminase (AST) contents were changed after stress but not the 2.0 LMWSA group. The survival rate after stress in 1.0 LMWSA (85.23%) and 2.0 LMWSA (80.20%) treatments was significantly higher than the Control (62.05%). The survival rate after stress negatively correlated with GST and ALT, introducing them as potential biomarkers for cadmium exposure in whiteleg shrimp. Accordingly, the 2.0 LMWSA treatment had the best performance in the abovementioned parameters. As the linear relation was observed, supplementing more levels of LMWSA to reach a plateau is recommended.

## 1. Introduction

Freshwater scarcity has been one of the most significant barriers to aquaculture development [1]. As a result, mariculture and coastal aquaculture have emerged as promising candidates for providing food for the next century [2]. This type of aquaculture now accounts for more than 55% of total aquaculture output (122 million tonnes) in 2020 [1]. Whiteleg shrimp (*Litopenaeus vannamei*) was the most-produced shrimp species in 2020, with 5.8 million tonnes [1]. This shrimp species dominates the production of coastal aquaculture and is an important source of foreign exchange income in many developing countries [1]. One of the most important themes for transforming Asian aquaculture (which produces more than 88% of total production) is biosecurity and disease control. The best suggestion has been the stocking of postrave shrimp sourced from pathogen-free broodstock and using novel feed additives to increase animal resistance to challenges [1]. Therefore, any additive that can improve growth and shrimp resistance against any challenge can be steps toward aquaculture sustainability. There are many challenges in the continuous development of this species’ aquaculture, such as heavy metal pollution, most importantly, cadmium (Cd).

Cd pollution has become a serious global environmental concern since this metal can be easily accumulated in the food chain and aquaculture production tissues and eventually negatively affects human health [3]. This heavy metal is relatively more soluble than others, causing high mobility in the food webs, and can exist for a long term in an environment as it is non-biodegradable [4]. Many studies have reported that Cd pollution in shrimp tissues in different parts of the world, such as the Persian and Oman Gulfs [5,6], Turkey [7], Vietnam [8,9], Bangladesh [10], China [11,12], India [13,14], and Mexico [15,16]. In terms of aquaculture species, Cd can negatively affect shrimp health and growth [17,18,19]. Some other studies tested different supplements to alleviate Cd toxicity. For example, probiotics in Nile tilapia (*Oreochromis niloticus*) [20], turmeric (*Curcuma longa*), and black pepper (*Piper nigrum*) in African catfish (*Clarias gariepinus*) [21], chitosan, or vitamin C alone or in a combination in common carp (*Cyprinus carpio*) [22], Chinese parsley (*Coriandrum sativum*) in rainbow trout (*Oncorhynchus mykiss*) [23], and onion (*Allium cepa*) in Nile tilapia [24] declined this metal toxicity.

Alginic acid is a naturally occurring, edible polysaccharide found in brown algae, and its combination with sodium makes sodium alginate. Some studies have used this additive to improve growth and immunity in aquatic species as a prebiotic [25] and boost the antioxidant system [26,27,28,29,30,31,32]. Further, sodium alginate has been used to improve broodstock reproductive performance, larval survival [33], shelf life [34], and fish sperm preservation [35]. Lower molecular weight increases solubility and fermentation [36] and is potentially more accessible and digestible for aquatic species. Few studies tested low molecular weight sodium alginate (LMWSA) in fish to improve immune parameters, resistance against bacteria, and growth [25,37,38,39].

To the best of our knowledge, no study tested the LMWSA on shrimps; further, no study in aquatic species tested its alleviation effect on heavy metal toxicity. We hypothesised that this additive could improve shrimp response to heavy metal exposure by boosting shrimp health. Therefore, this study was designed to examine how LMWSA can improve growth performance, survival rate, body composition, antioxidant response, and haemolymph parameters of whiteleg shrimp.

## 2. Material and Methods

### 2.1. Animal Ethical Statement

The national ethical framework for animal research in Iran and its guidelines [40] that were adopted from the Declaration of Helsinki (1975) and the Society for Neuroscience Animal Care and Use guidelines (1998) approved this study [40] to optimise handling and minimise animal stress.

### 2.2. Experimental Diets

The LMWSA supplement for this experiment was provided from Thailand, and the process of making LMWSA was explained elsewhere [41]. The commercial diet (Beyza Feed Mill 21 (Beyza 21 Manufacturing Company, pellet size:1.8–2.2 mm)) was powdered. We added powdered LMWSA to diets in four dosages, including Control, 0.5 LMWSA (0.5 g LMWSA per kg diet), 1.0 LMWSA (1 g LMWSA per kg diet), and 2.0 LMWSA (2 g LMWSA per kg diet). These dosages were chosen based on earlier studies in fish [25,37,41], as no study has been done in shrimps. After the milled diets became homogeneous by adding warm water, the resulting mixture was compressed by a meat grinder (Electrokar EC-1, Tehran, Iran) to form pellets with a 2 mm diameter. Then, pellets were spread out on a tray and dried in an oven to ≥90% dry matter at 60 °C for 24–48 h. After drying, the feeds were packed in suitable packages and kept at 4 °C [2]. The chemical compositions of experimental diets are presented in Table 1.

### 2.3. Shrimp and Husbandry Trial

This experiment was done at the Laboratory of Aquatic Research (Persian Gulf University, Bushehr, Iran). Post larvae shrimp in stage 12 were purchased from a local farm and were fed with starter diets (Beyza Feed Mill 21 (Beyza 21 Manufacturing Company, Shiraz, Iran), pellet size: 1.8–2.2 mm)) for 20 days. Then, shrimp was distributed to experimental tanks and fed with a Control diet for two weeks. Two hundred and forty whiteleg shrimp (3.88 ± 0.25 g) were stocked into 12 tanks (300 L) (20 shrimp per tank, triplicate). Tanks were filled with filtered and disinfected (chlorine, 10 ppm) seawater (40 ± 0.6 ppt), and about twenty percent of water was exchanged daily. The average temperature and pH were 30.0 ± 1.1 °C, 7.5 ± 0.5, ammonia-nitrogen 0.09 ppm, and the natural photoperiod was applied. During eight weeks of the feeding trial, shrimp were fed with the diets three times a day (8:00, 13:00, and 18:00 h) at 3% of body weight. 

### 2.4. Growth Performance

At the end of the feeding trial, all shrimp were fasted for 24 h and were then anesthetised with ice-cold water. Growth and feeding performances were evaluated by the following parameters: 

Specific growth rate (SGR) (% day^−1^) = 100 × (Ln [mean final body weight] − Ln [mean initial body weight])/time (days)) 

Weight gain (WG) (%) = 100 × ([mean final body weight − mean initial body weight]/mean initial body weight) 

Daily weight (g/day) = (mean final weight (g) − mean initial weight (g))/time (day)

Feed conversion ratio (FCR) = dry feed intake (g)/wet weight gain (g) 

Survival was calculated as follows: 

Survival (%) = 100 × (final shrimp number/initial shrimp number)

### 2.5. Biochemical Composition Analysis

The analysis of the proximate composition of shrimp was performed using the AOAC standard methods [42]. Briefly, the dry matter was measured gravimetrically after oven drying of homogenized samples for 24 h at 105 °C (AMB50; ADAM, Milton Keynes, UK). Crude protein (N × 6.25) was determined by the Kjeldahl procedure using an automatic Kjeldahl system (BÜCHI, Auto-Kjeldahl K-370; Flawil, Switzerland). Crude lipid was determined by ether extraction using Soxhlet (Barnstead/Electrothermal, Knutsford, UK), and ash content was determined after incineration in a muffle furnace (Finetech, Shin Saeng Scientific, Paju-si, Gyeonggi-do, Republic of Korea) at 550 °C for 6 h.

### 2.6. Haemolymph Collection

Seven shrimp from each tank were sampled to measure antioxidant parameters and serological enzymes. Hemolymph was collected directly from the cardiac sinus of shrimp with sterile syringes and transferred to centrifugal tubes on ice. Tubes were maintained at 4 °C overnight and then centrifuged at 1500× *g* rpm for 10 min at 4 °C. The hemolymph supernatant serum was collected and used for further analysis. 

### 2.7. Antioxidant Enzyme Activity Malondialdehyde Evaluation and Serological Enzymes

For evaluating the activity of liver antioxidant enzymes, hepatopancreas was quickly dissected and washed in ice-cold phosphate buffer (pH = 7.4) and immediately frozen in liquid nitrogen, then stored at −80 °C until homogenate preparation. Hepatopancreases were homogenised in ice-cold 100 mM phosphate buffer by using a homogeniser for 30–45 s. The tube was then centrifuged at 12,000× *g* for 30 min at 4 °C. Supernatants were collected and stored at −80 °C [43]. Glutathione S-transferase (GST) was measured by the absorbance increase at 340 nm, resulting from the conjugation of reduced glutathione (GSH) and 1- chloro-2,4-dinitrobenzene as described by [44]. The formation of malondialdehyde (MDA) was determined via the thiobarbituric acid method [45] with some modification. Briefly, to 100 µL homogenate, we added 1400 µL of 15% trichloroacetic acid dissolved in hydrochloric acid (0.25 N) and then added 14 µL of 2% butylated hydroxytoluene in methanol and mixed well. The mixture was heated in a 100 °C water bath for 15 min, then cooled to room temperature and centrifuged at 12,000× *g* for 5 min. Absorbance was measured in the supernatant at 532 nm, and we calculated the MDA concentration in samples based on the standard curve. Results expressed as nanomoles MDA formed per milligram protein. The GSH values were measured with the Ellman method [46]. For each sample, 100 μL of the sample was mixed with dithiobisnitrobenzoate (DTNB) and PBS. After an incubation of 5 min, the absorbance was read at 412 nm. The value of GSH was expressed as nmol/mg protein. The concentrations of haemolymph activities of aspartate aminotransferase (AST) and alanine transaminase (ALT) were determined using an automatic microplate reader (Synergy 2 Biorad) and Pars Azmun kit (Tehran, Iran).

### 2.8. Statistical Analysis

This experiment was conducted in a completely randomised design with four treatments and three replications. All data were analysed using SPSS 22.0 (SPSS Inc., Chicago, IL, USA). Normality and homogeneity of variance were tested initially using the Kolmogorov–Smirnov and Levene tests, respectively. For providing a comprehensive analysis, we applied orthogonal polynomial contrasts to determine if the LMWAS level had linear and/or quadratic relations with measured parameters [47]. Further, two-way ANOVA was done with the effect of diet and stress. When the interaction was significant, we unpacked the original data. When the interaction was not significant, we compared the main effects in pooled data). The data before and after stress was compared with an independent sample *t*-test. Data are presented as means ± standard deviation, and differences were considered to be significant at *p* < 0.05.

### 2.9. Cd Challenge Test

After eight weeks, ten shrimp from each experimental tank was distributed to Aquarium (40 L) for the challenge test. Acute toxicity of Cd as medium lethal concentration (LC50) values for whiteleg shrimp after 48 h was 1.30 mg/L [48]. For this experiment, we selected 1 mg/L as the Cd level for 48 h challenge. The solutions of metal were prepared with CdCl_2_ (Sigma-Aldrich Co., St. Louis, MI, USA) dissolved in distilled water to obtain a stock of 1 mg/L solution. The experimental metal mixture solution was obtained by adding the appropriate volume of each stock solution to the tanks. After finishing the exposure, hemolymph taken from the ventral sinus of five shrimp/aquariums was sampled with a 1 mL sterile syringe for further analysis. Shrimp were not fed during exposure. Shrimp survival percentages were recorded daily, and dead organisms were immediately removed from the aquaria.

## 3. Result

### 3.1. Growth Performance, Survival Rate, and Proximate Composition

The FCR was significantly higher in the control group than in those fed the 2 g LMWSA/kg diets. Results show that FCR has a linear relation with LMWSA levels in the diets (Table 2) (*p* < 0.05). There was no significant difference in growth performance and survival rate among groups. Results indicated that ash content had a quadratic relation with LMWSA levels in diets (*p* < 0.05) (Table 3). However, there was no significant difference in protein, lipid, and moisture contents among groups.

After Cd stress, the control group had the most mortality rate (*p* < 0.05), and there was a quadradic relation between LMWSA level and survival rate (Table 4).

### 3.2. Antioxidant Activities

There was a linear and quadratic relationship between GST and LWMSA levels, and with increasing its content in diets, GST went up (*p* < 0.05) (Figure 1). The results of two-way ANOVA indicated that the effect of stress on GSH was significant, and GSH decreased by stress (Table 5). The effect of diet on GSH was significant, and those fed dietary 2.0 LWMSA had higher levels compared to 0.5 LWMSA and control treatments. The interaction effect for MDA and GST was significant, and the original data was unpacked (Figure 1). Before stress, there was no significant difference in MDA levels. After stress, all parameters were changed, and GST and MDA decreased with increasing levels of LWMSA in diets (*p* < 0.05). The most important results were that the Control group had significantly higher values of GST and MDA parameters after stress than before, while the same results were not observed for shrimp-fed dietary LWMSA in dosages more than 0.1% (*p* < 0.05).

### 3.3. Haemolymph Enzymes

The results of two-way ANOVA indicated that the effect of stress on AST was significant, and this parameter increased by stress. The interaction effect for ALT was significant, and the original data was unpacked (Figure 2). Results indicate before stress, there were no significant differences in hepatopancreas ALT enzyme activities. However, after stress, whiteleg shrimp fed dietary LMWSA had a lower value of these enzymes compared to the Control (*p* < 0.05) (Figure 2). With stress, this parameter had higher values in the control and 0.5 LWMSA groups but not others.

## 4. Discussion

### 4.1. Growth Performance, SurvivalR, and Proximate Composition

The growing demand for novel supplements in aquaculture has attracted the attention of researchers to find alternative additives. Low molecular weight polysaccharides have recently been introduced as novel prebiotics [49,50]. Many studies on plant polysaccharides found that low molecular mass or hydrolysed oligosaccharides improved colonic persistence and increased fermentability by gut microflora [51,52,53,54].

Although several studies investigated the effect of sodium alginate on improved growth, immunity, and health of shrimps [28,29,30,33,55,56], no investigation tested the effect of the size of this supplement on animal performance. In the few studies available in aquaculture, tilapia-fed dietary LMWSA had higher growth performance, immunity, and resistance against bacterial challenge compared to the control [41]. The result of the present study indicated that whiteleg shrimp FCR has a linear relation with LMWSA levels in the diets (Table 2) (*p* < 0.05); therefore, more levels should be tested. The highest level in this study was 0.2%, while in tilapia, 0.3% of this supplement positively affected growth. However, the growth and SGR of whiteleg shrimp fed dietary LMWSA were not positively affected by this supplement. The reason might be that different species respond differently, and higher dosages should be tested to reach a plateau. Another possible hypothesis is that probably digestive system and microbiota of shrimp were not improved by changing the molecular weight. Unlike this study, some works showed improved growth by feeding animals with normal sodium alginate [56]. More research is required to determine the optimum dosage of this supplement in aquatic species.

There was no significant difference in survival rate among groups showing that LMWSA did not positively or negatively affect this parameter. Additionally, it can be said that shrimp were farmed in good condition, and the survival rate was higher than 89% in all treatments. After Cd stress, there was a quadradic relation between LMWSA level and survival rate; and the Control group had the most mortality rate (*p* < 0.05). This result clearly shows that this supplement improved whiteleg shrimp’s ability to tackle Cd stress. Similarly, other studies indicated that adding this supplement to tilapia diets improved the post-challenge survival rate against *Streptococcus agalactiae* [41]. In the present study, the higher survival rate might be due to the improvement of antioxidant defenses by the LMWSA. Similarly, sodium alginate improved the survival rate of common carp against Edwardsiella tarda infection [57].

While both internal (age, gender, and size) and external factors (water quality, season, and geographical location) affect the proximate body composition of aquatic species, the diet is most likely responsible for most of the changes [58]. The results of this study indicated that ash content had a quadratic relation with LMWSA level (*p* < 0.05) (Table 3). However, there was no significant difference in protein, lipid, and moisture contents. No change in the proximate composition of Malaysian Mahseer (*Tor tambroides*) with feeding sodium alginate was observed [32]. Decreasing protein, lipid, and ash contents by feeding tilapia with sodium alginate was also reported [59]. Further, when sea bream (*Sparus aurata*) was fed sodium alginate, lipid contents in the body were elevated. As was observed, there was a wide variety of responses in different species with feeding sodium alginate, and it is hard to make any solid conclusion. 

### 4.2. Antioxidant Activities

Measuring parameters such as MDA, GSH, and GST can be reliable markers of the antioxidant system and eventually shows the health status of animals. The Scopus database shows that more than 1900 articles have used these parameters to monitor aquatic species’ health in the last ten years. Imbalanced oxidative activities can bring superoxide and H_2_O_2_ radical damage, which antioxidant enzymes protect cells from occurring [60]. Cd toxicity causes oxidative stress in aquatic species [18,61,62]; therefore, measuring this parameter helps to understand how shrimp was affected by this stress. There is no study on shrimp related to the effect of sodium alginate on antioxidant parameters. However, when Asian sea bass (*Lates calcarifer*) were fed a 1% LMWSA diet, GST and MDA levels were significantly increased [26], reflecting improvement in the general health status of fish. Previous studies proved that most prebiotics stimulates the synthesis of the antioxidant enzymes such as glutathione. For example, crude polysaccharides [63], *Ganoderma lucidum* polysaccharides [64] in shrimps, and galactooligosaccharide in rainbow trout [65] resulted in this improvement. The many prebiotics’ effects on the antioxidant system were reviewed elsewhere [66,67,68,69]. The results of current data align with those studies and show that this supplement improved the antioxidant system of shrimp and caused less change in this system after Cd stress. The improved antioxidant system was in line with a higher survival rate in 1.0 LMWSA and 2.0 LMWSA groups (Table 4). Sodium alginate is a well-known strong antioxidant [70,71] and we observed this effect clearly in our study. More studies are required on shrimp to illustrate the potential improvement of the antioxidant system by LMWSA.

### 4.3. Haemolymph Enzymes

Serological enzymes such as AST and ALT are frequently examined to monitor the physiological status of aquatic species under different stressful or nutritional situations. Consistent with the growth data, there was no specific trend or change in ALT and AST enzymes, illustrating that the shrimp was in good condition. When shrimp were exposed to Cd stress, ALT and AST levels in the Control group elevated, but not in other treatments showing that those fed supplements had more stability in these enzymes. The survival rate in the control group was lower as well, and it can be hypothesised that if animals can control and have fewer changes in the antioxidant system and serological enzymes, they are more likely to able to cope with stress better and have a higher survival rate as was observed in LMWSA groups. More studies are required in this area to properly understand how liver enzymes can be changed by stress in shrimps. Many past studies have reported increased liver enzymes in response to various stresses [72,73,74,75,76]. More research is required to demonstrate the effect of Cd stress on the physiological status of shrimps.

### 4.4. Correlation between Measured Parameters

In the present study, a positive correlation between SGR and moisture contents (60%) and a negative correlation with ash (−68%) in the body was observed (*p* < 0.05) (Table 6). It can be hypothesised that the higher weight of the shrimp was due to higher moisture content. Further, FCR had negative correlations with LMWSA level (−58%) and GST (−70%) (*p* < 0.05). A negative correlation means lower FCR, showing that this supplement positively affected FCR. Further, the LMWSA level had positive correlations with GST (64%) and GSH after stress (71%); and a negative correlation with MDA after stress (−67%), GST after stress (−73%), and ALT after stress (−67%) (*p* < 0.05). It shows that LMWSA strongly affected the antioxidant system and serological enzymes in shrimp. These enzymes correlated with each other as well. For example, GSH after stress had negative relationships with MDA (−59%) and GST (−58%) (*p* < 0.05). Antioxidant parameters greatly correlated with serological enzymes showing that these two physiological systems are closely related to each other in whiteleg shrimp. In this way, GST after Cd stress had a positive correlation with ALT (84%) and AST (72%) (*p* < 0.05). MDA after Cd stress also had a positive relation with ALT (59%). Similarly, the same trend in these data was observed when whiteleg shrimp were fed Cd-polluted diets [77] and oxidised fish oil [78]. Interestingly, the survival rate after Cd stress had a negative correlation with enzymes such as MDA (−50), GST (−68%), ALT (−59%), and AST (−37%), which for GST and ALT were significant (*p* < 0.05). It is further evidence that these parameters can be indicators of survival rate after stress in whiteleg shrimp. The same trend was observed earlier, and stressed fish had higher values of these parameters [79,80].

## 5. Conclusions

Conclusively, LMWSA did not increase the growth rate but improved feed conversation efficiency. Regarding growth performance, the survival rate after Cd stress, antioxidant response, and serological enzymes, shrimps fed dietary 2.0 LMWSA had the best performance. No alteration in MDA, GST, ALT, and AST before and after Cd stress for the 1.0 LMWSA and 2.0 LMWSA groups caused shrimp to have a higher survival rate than the Control. After the Cd challenge, lower MDA, GST, ALT, and AST values were observed in the 2.0 LMWSA group. As there was a linear relationship between these parameters and LMWSA levels, supplementing more levels of this additive to diets is recommended to reach the optimum level. Further, the effect of LMWSA on Cd bioaccumulation in shrimp should be tested.

## Figures and Tables

**Figure 1 animals-13-01805-f001:**
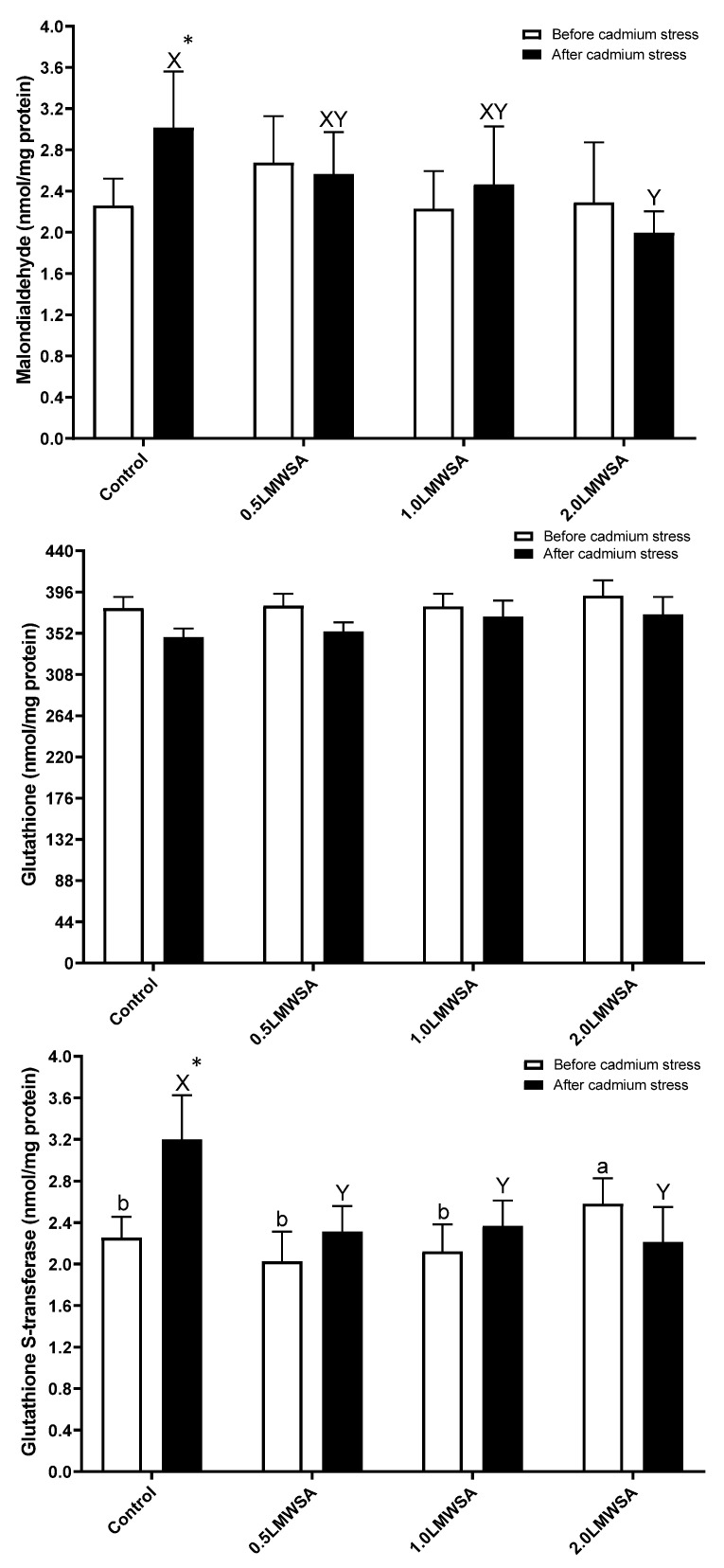
Antioxidant parameters in the hepatopancreas of whiteleg shrimp fed experimental diets containing different levels of low molecular weight sodium alginate (LMWSA). Letters a and b indicate significant differences in treatment before and X and Y after Cd stress, according to Duncan multiple range tests (*p* < 0.05). Asterisk shows a significant difference in each group before and after Cd stress via independent sample *t*-test (*p* < 0.05).

**Figure 2 animals-13-01805-f002:**
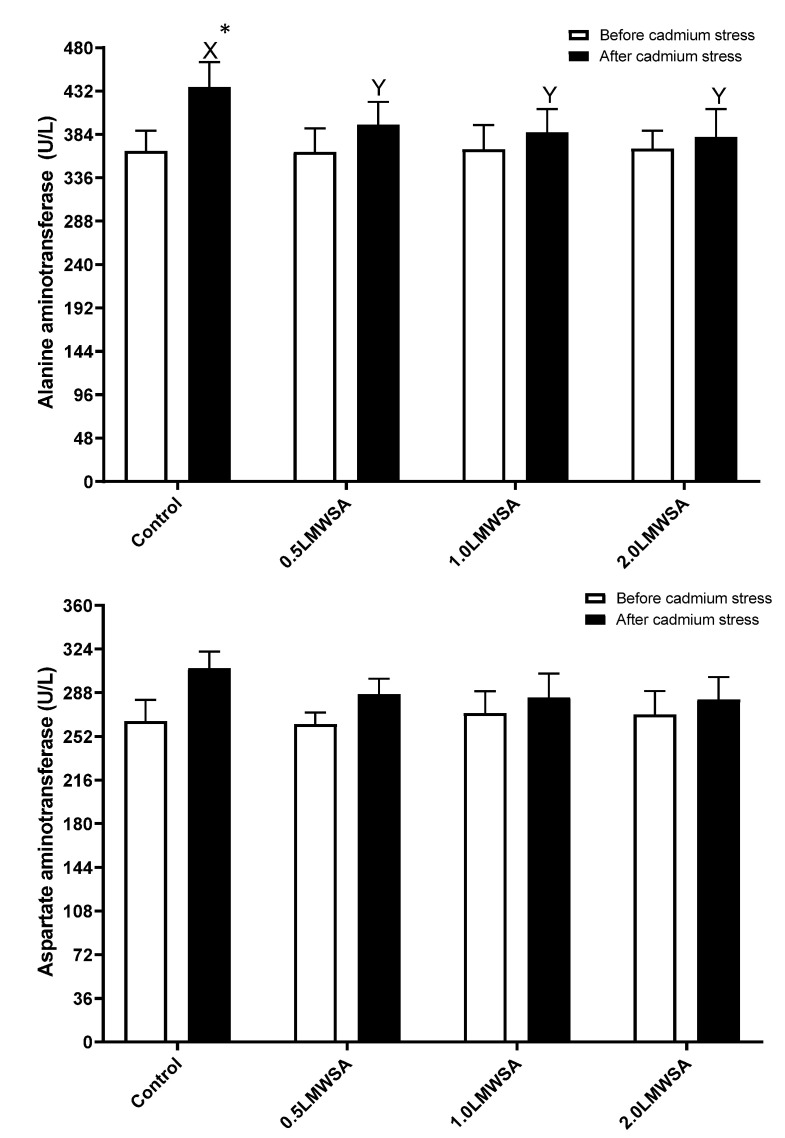
Serological enzymes in haemolymph of whiteleg shrimp fed experimental diets containing different levels of low molecular weight sodium alginate (LMWSA). Letters X and Y indicate significant differences in treatments after Cd stress, according to Duncan multiple range tests (*p* < 0.05). Asterisk shows a significant difference in each group before and after Cd stress via independent sample *t*-test (*p* < 0.05).

**Table 1 animals-13-01805-t001:** Proximate composition of the formulated feed.

Proximal Composition	(%)
Crude protein	40
Crude fat	9.2
Moisture	7.5
Crude ash	10.5
Crude fibre	3.1
Energy (kJ/g)	18.59
Nitrogen-free extract	37.2

**Table 2 animals-13-01805-t002:** Growth and feeding performances of whiteleg shrimp (*Litopenaeus vannamei*) fed low molecular weight sodium alginate (LMWSA) at different inclusion levels for eight weeks.

Parameters	Control	0.5 LMWSA	1.0 LMWSA	2.0 LMWSA
Initial weight (g)	4.00 ± 0.08	3.75 ± 0.38	3.81 ± 0.34	3.94 ± 0.13
Final weight (g)	14.75 ± 0.81	14.48 ± 0.43	13.60 ± 0.26	15.16 ± 0.98
Final length (cm)	11.83 ± 0.52	11.98 ± 0.25	11.88 ± 0.30	11.87 ± 0.46
SGR (%/day)	2.33 ± 0.06	2.42 ± 0.22	2.28 ± 0.15	2.40 ± 0.08
W.G. (g)	10.75 ± 0.73	10.73 ± 0.73	9.79 ± 0.35	11.22 ± 0.89
W.G. (%)	268.66 ± 12.59	289.32 ± 47.17	258.98 ± 29.24	284.20 ± 17.94
Feed consumption (g)	13.70 ± 0.61 a	14.49 ± 0.22 a	13.64 ± 0.27 a	12.28 ± 0.59 b
FCR	1.28 ± 0.04 a	1.36 ± 0.11 a	1.39 ± 0.03 a	1.10 ± 0.05 b
Survival (%)	91.67 ± 7.64	91.67 ± 10.41	91.67 ± 7.64	90.00 ± 8.66
Survival after Cd stress	62.05 ± 7.72	77.97 ± 6.61	85.23 ± 4.16	80.20 ± 9.31

Data in a row assigned with different letters indicate significant differences (*p* < 0.05). Values are reported as mean ± S.D.

**Table 3 animals-13-01805-t003:** Whole Body biochemistry composition of whiteleg shrimp (*Litopenaeus vannamei*) fed low molecular weight sodium alginate (LMWSA) at different inclusion levels for eight weeks.

Parameters	Control	0.5 LMWSA	1.0 LMWSA	2.0 LMWSA
Moisture (%)	71.01 ± 1.13	71.00 ± 1.04	71.45 ± 1.33	72.16 ± 0.79
Protein (%)	17.04 ± 0.10	16.98 ± 0.12	17.04 ± 0.10	16.63 ± 0.21
Fat (%)	1.45 ± 0.11	1.46 ± 0.21	1.49 ± 0.29	1.57 ± 0.18
Ash (%)	6.98 ± 0.09	6.63 ± 0.14	6.91 ± 0.11	6.86 ± 0.13

Values are reported as mean ± SD.

**Table 4 animals-13-01805-t004:** Report of SPSS for investigated parameters in this study that *p*-value of polynomial contrasts; linear or/and quadratic relation or ANOVA has been significant (*p* < 0.05). The NA means that the interaction effect of two-way ANOVA for this parameter was not significant, and pooled data was measured (Table 5).

Parameters	*p*-Value		
	ANOVA	Linear	Quadratic
Survival after stress	0.101	0.058	0.006
Feed conversion ratio	0.002	0.013	0.049
Ash	0.964	0.512	0.031
GST	0.018	0.023	0.006
MDA-challenege	0.001	0.003	0.009
GSH-challenege	NA	0.018	0.020
GST-challenege	0.002	0.008	0.001
ALT-challenege	0.007	0.026	0.009
AST-challenege	NA	0.162	0.048

**Table 5 animals-13-01805-t005:** The result of two-way ANOVA (diet*stress) for measured parameters before and after cadmium stress. Only significant parameters and those in which the interaction was not significant were reported. The parameters with the significant interaction effect (MDA, GST, and ALT) (diet*stress) were unpacked in the figures. The letters a and b indicated significant differences among groups based on the Duncan multiple range tests.

	Diet Effect	Stress Effect	Interaction	Before Stress	After Stress	Control	0.5 LMWSA	1.0 LMWSA	2.0 LMWSA
	*p* Value			Main Effects					
GSH	0.012	0.001	0.316	383.2	360.8	363.2 b	367.6 b	375.2 ab	382.0 a
AST	0.325	0.001	0.085	266.9	290.2				

**Table 6 animals-13-01805-t006:** Correlation between investigated parameters in whiteleg shrimp fed diets contained different levels of LMWSA for eight weeks.

	LMWSA Level	Survival-Challenge	FCR	SGR	GST	ALT	MDA-Challenge	GSH-Challenge	GST-Challenge	ALT-Challenge	AST-Challenge	Moisture	Ash
**LMWSA level**	**1.00**	0.56	−0.58 *	0.11	0.64 *	0.12	−0.67 *	0.71 **	−0.73 **	−0.67 *	−0.51	0.45	−0.23
**Survival-challenge**	0.56	**1.00**	0.23	−0.25	0.11	0.19	−0.50	0.56	−0.68 *	−0.59 *	−0.37	0.20	0.02
**FCR**	−0.58 *	0.23	**1.00**	−0.51	−0.70 *	0.15	0.24	−0.24	0.16	0.17	0.19	−0.48	0.03
**SGR**	0.11	−0.25	−0.51	**1.00**	−0.05	−0.59 *	−0.16	0.14	−0.34	−0.23	−0.41	0.60 *	−0.68 *
**GST**	0.64 *	0.11	−0.70 *	−0.05	**1.00**	0.12	−0.54	0.31	−0.05	−0.14	0.17	0.25	0.39
**ALT**	0.12	0.19	0.15	−0.59 *	0.12	**1.00**	0.21	−0.05	−0.03	−0.06	0.16	−0.19	0.34
**MDA-challenge**	−0.67 *	−0.50	0.24	−0.16	−0.54	0.21	**1.00**	−0.59 *	0.55	0.59 *	0.23	−0.25	0.22
**GSH-challenge**	0.71 **	0.56	−0.24	0.14	0.31	−0.05	−0.59 *	**1.00**	−0.58 *	−0.54	−0.47	0.51	−0.11
**GST-challenge**	−0.73 **	−0.68 *	0.16	−0.34	−0.05	−0.03	0.55	−0.58 *	**1.00**	0.84 **	0.72 **	−0.41	0.54
**ALT-challenge**	−0.67 *	−0.59 *	0.17	−0.23	−0.14	−0.06	0.59 *	−0.54	0.84 **	**1.00**	0.63 *	−0.14	0.25
**AST-challenge**	−0.51	−0.37	0.19	−0.41	0.17	0.16	0.23	−0.47	0.72 **	0.63 *	**1.00**	−0.46	0.38
**Moisture**	0.45	0.20	−0.48	0.60 *	0.25	−0.19	−0.25	0.51	−0.41	−0.14	−0.46	**1.00**	−0.28
**Ash**	−0.23	0.02	0.03	−0.68 *	0.39	0.34	0.22	−0.11	0.54	0.25	0.38	−0.28	**1.00**

SGR: specific growth rate; FCR: feed conversion ratio; GST: glutathione S-transferase; ALT: alanine transaminase; MDA: malondialdehyde; GSH: glutathione; AST: aspartate transaminase; * Correlation is significant at the 0.05 level (2-tailed); ** Correlation is significant at the 0.01 level (2-tailed).

## Data Availability

All data from this study are available upon request to the corresponding author.

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
