# Peer review of "Does Dietary Sodium Alginate with Low Molecular Weight Affect Growth, Antioxidant System, and Haemolymph Parameters and Alleviate Cadmium Stress in Whiteleg Shrimp (Litopenaeus vannamei)?"

_animals, 2023, doi:10.3390/ani13111805_

Round 1

Reviewer 1 Report

Very interesting and well subtantiated research which merits publication. However, it would be equally interesting to further investigate the effect of sodium alginate on the bio-accumulation of Cd in shrimp. This is equally important for the health status of aquaculture products in relation to human health. 

Author Response

Response: Many thanks for the comment, and you are right. Bioaccumulation of Cd will be a direct consequence and it would be interesting to see whether sodium alginate can reduce the heavy metal bioaccumulation or not. We will consider this point in our future studies.

Reviewer 2 Report

attached in report

Author Response

Title: Effects of Dietary Sodium Alginate with Low Molecular Weight on Growth, Body Composition, Antioxidant System, and Haemolymph Parameters of Whiteleg Shrimp (Litopenaeus vannamei) before and after Cadmium Challenge.
The title looks like thesis title; it must be refined as per the contents. Better it should end with
question marks.

Response: It is revised as you recommended.

Simple summary has written very causal. Author even did not maintain the simple grammatical
rules.

Response: It is revised as you recommended; it has now been updated. Apologies for this mistake; the wrong test was put in this section.

Abstract:
Four doses statistically not recommend and out of that author also found out optimum dose. It’s
not practically accepted from statistical point of view.

Response: We used both ANOVA and polynomial regression for this study. Please check the statistical analysis section. By doing this, we can benefit from both ANOVA and regression, as many researchers have used this approach.

Introduction
Line 44-45 instead of limitations, it should be replaced with scarcity for better interpretation.

Response: It is revised as you recommended.

2.1. Ethics statement
Authors are suggested to provide the name of approval authority and number of the application
for animal ethical. And the heading to be replaced as animal ethical statement.

Response: It is revised as you recommended. It was not any specific number to report, unlike animal ethic process in other parts of the world.

Table 1. Proximate composition of the formulated feed.
Why some values are represented as replicates value and others are single value and I suggest
authors to put single value.

Response: It is revised as you recommended.

2.4. Growth performance
What anesthesia authors have used. As mentioned only simple cold water serve the purpose.
Need to recheck.

Response: We used only cold water as it is shrimp, which could serve the purpose. If it was fish, of course, we had to use anesthesia.

Author wrongly written the some formulae i.e for condition factor what is the
unit?????????????.

Response: It is revised as you recommended. Condition factor is not a good parameter, and we deleted it.

Gram abbreviation is g not gr so author need to replace the same throughout the MS.

Response: It is revised as you recommended.

In table foot should indicate the sample size or replicates.

Response: Apologies, we did not understand this comment. Can you please explain more about it in detail?

For other formula they could give some references. I suggest putting following
references in this section. With preliminary datas author can find out some more indices
for supporting the results. Author can follows following articles for the same.
Manish Jayant, Narottam Prasad Sahu, Ashutosh Dharmendra Deo, Subodh Gupta, Kooloth
Valappil Rajendran, Chetan Kumar Garg, Dharmendra Kumar Meena, Minal Sheshrao Wagde,
Effective valorization of bio-processed castor kernel meal based fish feed supplements
concomitant with oil extraction processing industry: A prolific way towards greening of
landscaping/environment, Environmental Technology & Innovation,Volume
21,2021,101320,ISSN 2352-1864.
Geetanjali Yadav, Dharmendra Kumar Meena, Amiya Kumar Sahoo, Basanta Kumar Das,
Ramkrishna Sen,Effective valorization of microalgal biomass for the production of nutritional
fish-feed supplements,Journal of Cleaner Production,Volume 243,2020,118697,ISSN 0959-
6526.

Response: Thanks for the suggestion and we added an appropriate reference for the formula (reference book).

Discussion and results missing the cohesion so it needs a significant revision with previous
studies as references.

Response: It is revised as you recommended.

Conclusion needs refinement.

Response: It is revised as you recommended.

Reviewer 3 Report

This study investigated the effects of dietary sodium alginate with low molecular weight on the growth, body composition, antioxidant capability, and haemolymph function of whiteleg shrimp (Litopenaeus vannamei). The topic is of interest for aquaculturists. Generally, the manuscript was arranged in a straight-forward way, and is easy to understand. The experimental design is scientifically sound. The results obtained could advance the development of functional feed additives for this species. However, there are some concerns that should be addressed by the authors. Please refer to the following comments.

Major ones:

1.      The authors stated that alginic acid is often used to improve the growth and immunity in aquatic species as a prebiotic (line 80). However, no immunological parameters are investigated in this study.

2.      The following parameters are recommended to be provided to facilitate the understanding of the results: 1) feed consumption and the hepatosomatic index; 2) the ones related to stress response in haemolymph, like the concentrations of cortisol, glucose and lactate.

3.      Statistical analysis method is not fully correct. For example, two-way ANOVA should be adopted to analyze the data before and after Cd stress. Using an independent sample T-test only is not correct. In addition, the survival data presented in Table 2 should be reanalyzed taking into consideration the effects of Cd stress and dietary treatment.

4.      Why the polynomial contrast analysis was only conducted for the parameters listed in Table 4, not others?

5.      The M & M section was not well organized. For example, the “2.7. Antioxidant enzyme activity malondialdehyde evaluation and serological enzymes” was followed by the “2.10. Statistical analysis”. In addition, another 2.7 section (namely the Cd challenge test) was also presented. Please double check this section to eliminate these mistakes.  

6.      In the M & M section, the procedures for measuring the GSH content were detailed. However, in Fig. 1, the glutathione peroxidase content was presented instead of GSH. In addition, the GST was measured for activity or content, since the unit is “nmol/mg protein” not “U/mg protein”?

7.      The reference section should be double checked to avoid mistakes. For example: 1) no volume, issue and page information is available for reference 1; 2) the journal names for reference 8 and 29 were not correctly presented; 3) no page information is available for reference 25, etc.

Minor ones:

1.      The unit of sodium alginate should be consistent throughout the manuscript. In this case, both % and g/Kg was adopted in the text.

2.      Lines 173-176, the description concerning the function of GSH should be deleted.

3.      The unit for GPX and GST should be changed to U/IU. In addition, the unit of AST and ALT should be “U/L” not “u/L” .

4.      The unit of liter should be consistent throughout the manuscript. In this case, both l and L was adopted in the text.

5.      The font size of growth performance in line 210 is not correct. In addition, the letter P (concerning the significant value) should be italic throughout the manuscript.   

Author Response

This study investigated the effects of dietary sodium alginate with low molecular weight on the growth, body composition, antioxidant capability, and haemolymph function of whiteleg shrimp (Litopenaeus vannamei). The topic is of interest for aquaculturists. Generally, the manuscript was arranged in a straight-forward way, and is easy to understand. The experimental design is scientifically sound. The results obtained could advance the development of functional feed additives for this species. However, there are some concerns that should be addressed by the authors. Please refer to the following comments.

Major ones:

  1. The authors stated that alginic acid is often used to improve the growth and immunity in aquatic species as a prebiotic (line 80). However, no immunological parameters are investigated in this study.

Response: We wanted to review the effects of this supplement on aquatic species as it is an introduction section. The immune system was one of them, but we did not mean to only mention the investigated parameters. In the discussion section, we only compared our data with the studies that similarly tested measured parameters in the current research.

  1. The following parameters are recommended to be provided to facilitate the understanding of the results: 1) feed consumption and the hepatosomatic index; 2) the ones related to stress response in haemolymph, like the concentrations of cortisol, glucose and lactate.

Response: Thanks for this comment. However, shrimp has no liver, and we do not have hepatosomatic index in shrimp. Feed consumption was added to Table 2. We do not have enough resources and funds to measure these parameters. We will consider this point in future studies.

  1. Statistical analysis method is not fully correct. For example, two-way ANOVA should be adopted to analyse the data before and after Cd stress. Using an independent sample T-test only is not correct. In addition, the survival data presented in Table 2 should be reanalyzed taking into consideration the effects of Cd stress and dietary treatment.

Response: It is revised as you recommended. We also updated the figures and results based on the new analysis.

  1. Why the polynomial contrast analysis was only conducted for the parameters listed in Table 4, not others?

Response: As it is mentioned in the title of the table, only significant parameters were reported here. Otherwise, we did for all parameters for this study.

  1. The M & M section was not well organized. For example, the “2.7. Antioxidant enzyme activity malondialdehyde evaluation and serological enzymes” was followed by the “2.10. Statistical analysis”. In addition, another 2.7 section (namely the Cd challenge test) was also presented. Please double check this section to eliminate these mistakes.

Response: It is revised as you recommended; I apologise for the mistake.

  1. In the M & M section, the procedures for measuring the GSH content were detailed. However, in Fig. 1, the glutathione peroxidase content was presented instead of GSH. In addition, the GST was measured for activity or content, since the unit is “nmol/mg protein” not “U/mg protein”?

Response: It is revised as you recommended.

  1. The reference section should be double checked to avoid mistakes. For example: 1) no volume, issue and page information is available for reference 1; 2) the journal names for reference 8 and 29 were not correctly presented; 3) no page information is available for reference 25, etc.

Response: We will modify this in the last steps of acceptance as the list of references have been changed during the review process.

Minor ones:

  1. The unit of sodium alginate should be consistent throughout the manuscript. In this case, both % and g/Kg was adopted in the text.

Response: It is revised as you recommended.

  1. Lines 173-176, the description concerning the function of GSH should be deleted.

Response: It is revised as you recommended.

  1. The unit for GPX and GST should be changed to U/IU. In addition, the unit of AST and ALT should be “U/L” not “u/L” .

Response: For antioxidant parameters, the unit was for content, and we revised it in the MS. For ALT and AST, the unit was revised as recommended.

  1. The unit of liter should be consistent throughout the manuscript. In this case, both l and L was adopted in the text.

Response: It is revised as you recommended.

  1. The font size of growth performance in line 210 is not correct. In addition, the letter P (concerning the significant value) should be italic throughout the manuscript.

Response: It is revised as you recommended.

Many thanks for your time and effort in reviewing this MS.

Round 2

Reviewer 2 Report

article can be accepted now for publication 

Reviewer 3 Report

 All comments were responded appropriately by the authors with the manuscript revised accordingly. The manuscript can be accepted for publication.